# United Kingdom Research study into Ethnicity And COVID-19 outcomes in Healthcare workers (UK-REACH): a retrospective cohort study using linked routinely collected data, study protocol

Lucy Teece [iD],[1] Laura J Gray [iD],[1] Carl Melbourne [iD],[2] Chris Orton [iD],[3] David V Ford,[3] Christopher A Martin [iD],[4,5] David McAllister,[6] Kamlesh Khunti [iD],[7,8] Martin Tobin [iD],[2,9] Catherine John [iD],[2] Keith R Abrams [iD],[10] Manish Pareek [iD],[4,5] The UK-REACH Study Collaborative Group

For numbered affiliations see end of article.

**Correspondence to**
Manish Pareek;
manish.pareek@leicester.ac.uk

## ABSTRACT

**Introduction** COVID-19 has spread rapidly worldwide, causing significant morbidity and mortality. People from ethnic minorities, particularly those working in healthcare settings, have been disproportionately affected. Current evidence of the association between ethnicity and COVID-19 outcomes in people working in healthcare settings is insufficient to inform plans to address health inequalities.

**Methods and analysis** This study combines anonymised human resource databases with professional registration and National Health Service data sets to assess associations between ethnicity and COVID-19 diagnosis, hospitalisation and death in healthcare workers in the UK. Adverse COVID-19 outcomes will be assessed between 1 February 2020 (date following first confirmed COVID-19 case in UK) and study end date (31 January 2021), allowing 1-year of follow-up. Planned analyses include multivariable Poisson, logistic and flexible parametric time-to-event regression within each country, adjusting for core predictors, followed by meta-analysis of country-specific results to produce combined effect estimates for the UK. Mediation analysis methods will be explored to examine the direct, indirect and mediated interactive effects between ethnicity, occupational group and COVID-19 outcomes.

**Ethics and dissemination** Ethical approval for the UK-REACH programme has been obtained via the expedited HRA COVID-19 processes (REC ref: 20/HRA/4718, IRAS ID: 288316). Research information will be anonymised via the Secure Anonymised Information Linkage Databank before release to researchers. Study results will be submitted for publication in an open access peer-reviewed journal and made available on our dedicated website (https://uk-reach.org/).

**Trial registration number** ISRCTN11811602.

## INTRODUCTION

COVID-19 has been declared a global pandemic. Since the first reported cases in

## Strengths and limitations of this study

► First large UK-wide study assessing the associations between ethnicity and COVID-19 outcomes in people who work in healthcare settings.
► Ambitious linkage of multiple anonymised human resource and professional registration databases with National Health Service data sets, aiming to provide coverage of the whole of the UK.
► Broad and comprehensive analysis that aims to unpick the associations between ethnicity, occupational group and multiple COVID-19 outcomes while adjusting for core predictors.
► Access to highly detailed human resource databases (including Electronic Staff Record for England and Wales) enables granular analyses of ethnicity, occupation and COVID-19 exposures.
► While every effort has been made to identify as many healthcare workers as possible, the lack of a single UK-wide database capturing those actively working within healthcare settings is a key limitation that may reduce our ability to identify all healthcare workers across all regions.

December 2019, over 110.7 million cases and 2.4 million deaths have been reported worldwide.[1] Consistent evidence identifies advanced age, male sex, socioeconomic deprivation and chronic comorbidities as core predictors for COVID-19 adverse outcomes.[2–5]

The COVID-19 pandemic has disproportionately impacted historically disadvantaged populations.[6] In the UK, higher infection and mortality rates are observed in areas with greater deprivation[7] and among ethnic minorities.[8–12] This has prompted calls to prioritise the publication of data disaggregated by ethnicity and to conduct research which increases our understanding of risk in

these populations.[13] [14] Ethnicity is a complex construct which represents social, biological or genetic differences between populations.[15] Potential explanations for COVID-19-related health disparities between ethnic groups include socioeconomic, cultural, behavioural, biological or pathophysiological differences.[16] It is recommended that investigations into ethnicity-related health inequality should also consider interrelated factors such as deprivation, religion and pre-existing conditions. This approach serves to highlight the interplay between ethnic disparities and economic inequality[13] and disentangle the independent importance of these factors.[14]

Those working in healthcare settings, hereafter referred to as Healthcare Workers (HCWs), have a greater risk of SARS-CoV-2 exposure and transmission due to the number of COVID-19 cases requiring medical intervention.[17] A Scottish linkage study concluded over a sixth of hospitalised COVID-19 cases were HCWs or members of their households.[18] There is concern that HCWs from ethnic minority groups are at increased risk of adverse outcomes from COVID-19. However, the interplay between ethnicity and healthcare occupation and its impact on COVID-19 outcomes is unclear. Publicly available data indicate that individuals from ethnic minorities account for 65%–76% of deaths reported in clinical HCWs, despite contributing less than 20% of the National Health Service (NHS) workforce.[19] [20] We must understand differences in risks in these populations to protect the most vulnerable HCWs and maintain a functioning healthcare system.[18] However, the quality of data available to evaluate COVID-19 outcomes in HCWs, and the impact of ethnicity on these outcomes, remains poor and limited by small sample size.

The United Kingdom Research study into Ethnicity And COVID-19 diagnosis and outcomes in Healthcare workers (UK-REACH) programme will rapidly examine differences in COVID-19 diagnosis, clinical outcomes, professional practices and well-being among HCWs from different ethnicities. It will provide rapid evidence through five interlinked work packages; a large data linkage cohort study, a longitudinal cohort study, a legal/ethical analysis, a qualitative work stream and the development of a stakeholder working group. Here we describe the protocol for the large data linkage cohort study, which will bring together anonymised human resource (HR), professional registration and NHS data sets within a Trusted Research Environment.[21] This linked database will be used to assess the relationship between ethnicity and COVID-19 diagnosis, hospitalisation and death in HCWs.

## Objectives

Our primary aim is to determine whether COVID-19 diagnosis, hospitalisation and mortality rates differ between ethnic and occupational groups in HCWs. We will conduct both a broad analysis, encompassing all those registered as HCWs on 1 February 2020 in the UK, and a detailed substudy analysis in those actively working during the

COVID-19 pandemic, incorporating more granular ethnicity, occupation and exposure information.

Considering current evidence, we hypothesise that HCWs from ethnic minority backgrounds will have increased risk of COVID-19 diagnosis and adverse outcomes compared with their White counterparts regardless of occupation.

## METHODS
### Study design

We will use data from multiple linked electronic health record (EHR) databases to investigate the study hypotheses within a cohort of HCWs. The cohort study will begin on 1 February 2020 (date following first confirmed COVID-19 case in UK) and follow participants for 1 year, until 31 January 2021.

### UK healthcare data sources

For all residents in the UK, healthcare is free at the point of delivery as part of the Government-funded NHS. The NHS employs around 1.6 million people in hospital and community health services; over half are professionally qualified clinical staff (including doctors, nurses, midwives, technical and ambulance staff), the remainder support clinical staff and the NHS Infrastructure. However, identification of all HCWs in the UK is not straightforward. There is no single UK-wide database capturing these employees as the health systems in England, Scotland, Wales and Northern Ireland have diverged since political devolution in 1999. Further, many staff groups that provide NHS services, including general practitioners (GPs) and dentists, are not directly employed by the NHS.

Our primary data sources for identifying HCWs are the NHS HR and payroll databases, namely the Electronic Staff Record (ESR), capturing approximately 1.4 million staff paid through the NHS in England and Wales, and the Scottish Workforce Information Standard System (SWISS) including approximately 170 000 staff directly employed by NHS Scotland; as well as clinical staff, these databases include NHS paid ancillary and administrative services staff, and those in management roles. These HR databases cover NHS organisations including hospital trusts, ambulance services and clinical commissioning groups with comprehensive employment data during the cohort period of interest (table 1). We also aim to access HR data from non-NHS ancillary worker contractor companies to capture non-clinical staff and those working in facilities management and care coordination positions.

To enable wider coverage of the UK and capture workers in non-NHS sectors (eg, community and primary care), the HR data will be supplemented by health and social care regulators, where available. These include the General Medical Council (GMC), Nursing and Midwifery Council (NMC), General Dental Council (GDC), General Pharmaceutical Council (GPhC), General Optical Council (GOC), Pharmaceutical Society Northern Ireland (PSNI)

**Table 1** Coverage of human resources and professional registration databases utilised to identify UK healthcare worker cohort for the planned study

| Registry | Approximate size | Country coverage | Occupational coverage |
|---|---|---|---|
| Electronic Staff Record (ESR) | 1.4 million | England and Wales* | Staff directly employed by NHS England and Wales. |
| Scottish Workforce Information Standard System (SWISS) | 163 k | Scotland | Staff directly employed by NHS Scotland. |
| General Medical Council (GMC) | 336 k | England, Wales, Scotland and Northern Ireland | All doctors.† |
| Nursing & Midwifery Council (NMC) | 706 k | England, Wales, Scotland and Northern Ireland | Nurses, midwives, student nurses and nursing associates.‡ |
| General Dental Council (GDC) | 110 k | England, Wales, Scotland and Northern Ireland | Dentists, dental care professionals and dental practices.§ |
| General Optical Council (GOC) | 30 k | England, Wales, Scotland and Northern Ireland | Optometrists, dispensing opticians, student opticians. |
| General Pharmaceutical Council (GPhC) | 94 k | England, Wales and Scotland | Pharmacists and pharmacy technicians. |
| Pharmaceutical Society Northern Ireland (PSNI) | 3 k | Northern Ireland | Pharmacist and trainee pharmacists. |
| Health and Care Professions Council (HCPC) | 280 k | England, Wales, Scotland, Northern Ireland and international¶ | Art therapists, biomedical scientists, chiropodists, podiatrists, clinical scientists, dietitians, hearing aid dispensers, occupational therapists, operating department practitioners, orthoptists, paramedics, physiotherapists, practitioner psychologists, prosthetists, orthotists, radiographers, speech and language therapists. |

*Two foundation trusts in England do not currently use ESR.
†Including those in UK Foundation Year 1 and 2 posts, general practitioners and specialist consultants.
‡In England only.
§Dental care professionals include clinical dental technicians, dental hygienists, dental nursed, dental technicians, dental therapists, orthodontic therapists.
¶International professionals will not be included in this study as linkage to electronic health records, thus assessment of outcomes will not be possible.

and Health and Care Professions Council (HCPC). Where possible, we will obtain data for all HCWs registered to these regulators on the cohort inception date (1 February 2020). As these registries provide less information on specific occupational role and workplace location, it is not possible to identify those actively working during the study period. All included bodies provide UK-wide coverage, except pharmaceutical bodies GPhC and PSNI, which regulate Great Britain and Northern Ireland separately. Table 1 compares the coverage, inclusion, and data available from each registration database.

Given the differences in coverage and data across databases, the analysis will be twofold. First, a broad analysis will address the main objective using the full UK-wide cohort for any HCW employed or registered on 1 February 2020. Second, a detailed HR substudy will incorporate additional information on ethnicity, occupational role and potential for SARS-CoV-2 exposure (including whether an individual actively worked during the study period), and changes to these throughout the study period.

COVID-19 diagnosis, hospitalisation and mortality data will be gathered from several routinely collected EHR databases. We aim to obtain primary and secondary care data sets as well as specific intensive care audit data (such as Intensive Care National Audit & Research Centre data for England and Wales and Scottish Intensive Care Society Audit Group data for Scotland) and mortality information (Office for National Statistics, ONS) for each country. We also plan to incorporate specialised COVID-19 related databases, including COVID-19 surveillance data (COVID-19 Hospitalisation in England Surveillance System, CHESS), COVID-19 Symptom Study app data,[22] and Public Health England (PHE) Pillar 1 (within hospital) and Pillar 2 (community-based) testing data, where available. Additional data sets, such as PHE Pillars 3 and 4, will be sought and incorporated if obtained.

It may not be possible to access all outcome data sets for England, Wales, Scotland and Northern Ireland. Therefore, outcomes will be assessed separately in each country and results combined using meta-analysis methods to produce UK-wide estimates.

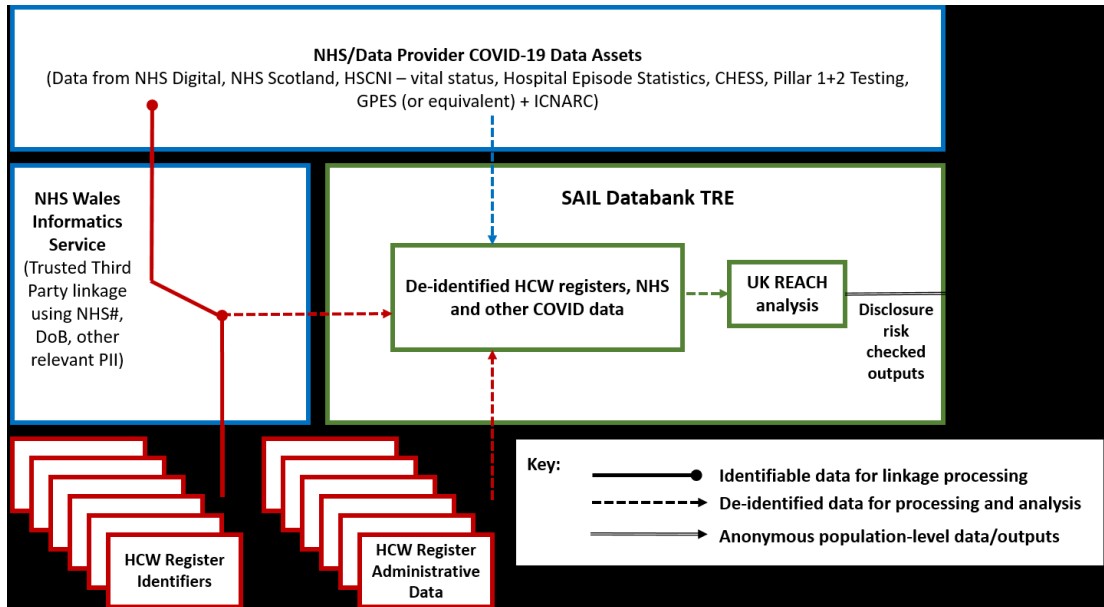

**Figure 1** Data flow and linkage of UK-REACH data sources. The above diagram and style was interpreted from an initial data flow diagram created and provided by Andy Boyd at the University of Bristol. It has been repurposed and amended to illustrate data flows specific to the UK-REACH project by Chris Orton at Swansea University. UK-REACH, United Kingdom Research study into Ethnicity And COVID-19 diagnosis and outcomes in Healthcare workers.

## Data linkage

Anonymised data will be stored at the Secure Anonymised Information Linkage (SAIL) databank.[21] SAIL works in partnership with a trusted third party, the NHS Wales Informatics Service (NWIS), to ensure processing of personal data is kept within the NHS and separate from the research environment. The process for anonymisation and linkage is depicted in figure 1.

The process to create the cohort of HCWs is as follows: Information from the HR and regulator databases will be separated into identifiable data and other administrative data. The identifiable data will be sent directly to NWIS and anonymised by encrypting identifiers into a single anonymous ID. Databases will be linked using this ID to identify the study cohort of HCWs and duplicate information from individuals present in more than one of the HR and regulator databases will be removed. Other administrative data will be sent directly to SAIL, who will use the anonymous study ID (subsequently removed to prohibit reverse engineering of re-identification) to match to the anonymised data.

The process for obtaining linked healthcare records for the cohort is as follows: NWIS will share the identifiers from the study cohort with the relevant health record providers, such as NHS Digital, NHS Scotland and the Northern Ireland Honest Broker Service, to establish linkage to required data sets. The health record providers then return the linked health records, including information for outcomes and predictors, following the same split-file process (identifiable data to NWIS, anonymised data to SAIL). Identifiable health record data will be kept entirely within the NHS. Linked anonymised records will be matched in SAIL using the anonymous study ID to

enable the project analysis. Once the linkage process is complete, the identifiable data will be deleted in accordance with the contractual arrangement between NWIS and SAIL.

We will develop a process to enable access to the UK-REACH data for other researchers once this study is complete. This process will engage with the above health record providers to establish onward sharing in partnership with the SAIL Databank.

## Study population

The cohort will include all adult (aged 16 years and over) HCWs captured by HR records or registered with at least one professional regulator on 1 February 2020, excluding those which linkage to EHRs is unsuccessful. HCWs with missing ethnicity, occupation, age, sex or postcode will be excluded from the main analysis. For the HR substudy, only those actively working will be included, thus excluding those on parental, sickness or disability leave for the duration of study period.

## Observation period

Follow-up commences on the index date (1 February 2020) (or date of return/commencement of employment for the HR substudy) and continue until either their death date (COVID-19 related or due to other causes) or the study end date (31 January 2021). For the HR substudy, those leaving NHS employment during the study period will be included up until this date, at which point they will be censored.

## Main exposures

The main exposure of interest is ethnicity, which is likely to be inconsistently recorded across databases. We will

**Table 2** ONS ethnicity groupings

| Broad ethnic categories | Ethnic groups |
|---|---|
| White | English/Welsh/Scottish/Northern Irish/British<br>Irish<br>Gypsy or Irish Traveller<br>Any other White background |
| Mixed/multiple ethnic groups | White and Black Caribbean<br>White and Black African<br>White and Asian<br>Any other mixed/multiple ethnic background |
| Asian/Asian British | Indian<br>Pakistani<br>Bangladeshi<br>Chinese<br>Any other Asian background |
| Black/African/Caribbean/Black British | African<br>Caribbean<br>Any other Black/African/Caribbean background |
| Other ethnic group | Arab<br>Any other ethnic group |

ONS, Office for National Statistics.

primarily assign ethnicity using staff HR records (ESR or SWISS) and registration databases (if not included/missing from HR records). If ethnicity is missing from these databases, linked EHRs will be assessed. Those with ethnicity recorded as 'not stated' or with completely missing ethnicity information will be excluded from the main analyses. Ethnicity will be categorised based on ONS groupings (table 2).

A more granular analysis will incorporate subdivisions of ethnicity (particularly within Asian/Asian British category) if a sufficient number of events are observed.

Occupational group (and its interaction with ethnicity) is also of primary interest. We will assign occupation for the main cohort using data recorded on 1 February 2020 from either HR records or data held by the regulating bodies and the following categories informed by version 17.1 of the NHS Occupational Code Manual (*NHS Digital*):

► Ambulance staff
► Administrations and estates staff
► Healthcare assistants and other support staff
► Medical staff
► Dental staff
► Nursing, midwifery and health visiting staff
► Nursing, midwifery and health visiting learners
► Scientific, therapeutic and technical staff
► Healthcare sciences
► General payments

A more granular analysis will incorporate subdivisions of occupational groups (particularly the Medical staff group) if a sufficient number of events are observed.

## Outcomes

The main outcomes are COVID-19 diagnosis (confirmed and suspected, confirmed only), hospitalisation (all, COVID-19 specific, ITU admission) and mortality (all-cause, COVID-19 specific). The definitions, data sources and a summary of planned analyses for these outcomes are provided in table 3: COVID-19 outcomes and definitions, identified via listed data sources, to be assessed in UK HCW cohort and within listed subgroups, via listed analysis approach in the planned study.

## Predictors

We will investigate and adjust for the following core predictors: age, sex, comorbidities and socioeconomic deprivation. A description of the definitions for these core predictors, and how they will be assessed, is provided in table 4.

Elevated body mass index (BMI), obesity and smoking are also considered key predictors for adverse COVID-19 outcomes;[9 23 24] however, are likely to be missing for large proportions of the identified cohort. Furthermore, missingness in EHR data is unlikely to be random, preventing standard imputation approaches. Thus, BMI and smoking will not be included as independent predictors in the primary study but instead in a sensitivity analysis exploring COVID-19 outcomes within HCWs where this information is available in primary or secondary care medical records, within 24 months prior to the study start date.

## HR sub-study

This sub-study will consider only those undertaking active employment in the NHS during the study observation period and will be conducted only in regions where data are available.

Additional ethnicity-related predictors include religious belief, nationality, immigration status and country of birth. Additional occupation predictors include primary area of work, role description, full/part time, patient/non-patient facing,[18] time in current role, organisation type, salary grade, length of service and seniority. Additional exposure-related predictors include absence information and total hours worked. Changes in these predictors will be captured throughout the study period. Time varying covariates will be incorporated to enable the assessment of the impact of changes in COVID-19 exposure over time.

## Patient and public involvement

We have been working with organisations that regulate and represent ethnic minority HCWs, including the GMC, NMC, Royal Colleges and professional associations such as the British Association of Physicians of Indian Origin. They were involved in identifying research questions and deciding study methodology, and are included as members of the study delivery team or collaborators. These organisations are part of the stakeholder group, meeting monthly to monitor study progress and outputs,

**Table 3** COVID-19 outcomes and definitions, identified via listed data sources, to be assessed in UK healthcare worker cohort and within listed subgroups, via listed analysis approach in the planned study

| Outcomes | | Definitions | Data sources | Subgroups | Analysis approach (output) |
|---|---|---|---|---|---|
| COVID-19 diagnosis | Confirmed COVID-19 diagnosis | A positive swab test for SARS-CoV-2 during the study period | PHE Pillar 1 (PHE laboratories or NHS hospital) and Pillar 2 (wider population) testing data[1] | | Poisson regression (IRR) |
| | Suspected COVID-19 diagnosis | Self-reporting of either a positive SARS-CoV-2 test or combination of symptoms shown to be predictive of positive testing[2] during the study period or suspected case SNOMED CT codes (NHS Digital) recorded during the study period | COVID-19 Symptom Study app and primary care data | | Poisson regression (IRR) |
| Hospitalisation | All hospitalisations | Any hospital admission recorded during the study period | Secondary care and CHESS data | Confirmed COVID-19 diagnosis | Logistic regression (OR) |
| | COVID-19 hospitalisations | hospital admissions with COVID-19-specific ICD-10 codes[3] during the study period or positive swab test for SARS-CoV-2 recorded during or within 28 days prior to admission | Secondary care, CHESS, PHE Pillar 1 (PHE labs or NHS hospital) and Pillar 2 (wider population) testing data[1] | Confirmed COVID-19 diagnosis | Logistic regression (OR) |
| | COVID-19 ITU admissions | Admissions to ITU during the study period | Secondary care, CHESS, and intensive care audit data | Confirmed COVID-19 diagnosis | Logistic regression (OR) |
| Mortality | All-cause mortality | Any death registration recorded during the study period | ONS mortality, primary care and secondary care data | Confirmed COVID-19 diagnosis COVID-19 hospitalisations COVID-19 ITU admissions | Flexible parametric time-to-event regression (HR) |
| | Mortality due to COVID-19 | any death registration with COVID-19-specific ICD-10 codes[3] recorded for the primary or secondary cause of death during the study period | ONS mortality, primary care and secondary care data | Confirmed COVID-19 diagnosis COVID-19 hospitalisations COVID-19 ITU admissions | Flexible parametric time-to-event regression (HR) |

*PHE Pillar 3 (serology testing for antibodies) and Pillar 4 (blood and swab testing) data will be incorporated if obtained.
†Symptoms include loss of smell or taste, fatigue, persistent cough and loss of appetite.[34]
‡COVID-19-specific ICD-10 codes are U07.1: 'COVID-19, virus identified' and U07.2: 'COVID-19, virus not identified'.
CHESS, COVID-19 Hospitalisation in England Surveillance System; ICD-10, International Classification of Diseases—10th revision; IRR, incidence rate ratio; ITU, Intensive Therapy Unit; NHS, National Health Service; ONS, Office for National Statistics ; PHE, Public Health England; SNOMED CT, Systematised Nomenclature of Medicine Clinical Terms.

**Table 4** Definitions for core predictors associated with risk of adverse COVID-19 outcomes

| Predictor | Definition |
|---|---|
| Age | A continuous measure of age (in years) at the index date (1 February 2020), calculated using date of birth as recorded in HR or professional registration databases. |
| Sex | A categorical measure (female/male/other) as recorded in HR or professional registration databases. |
| Comorbidities | A categorical measure counting the number of comorbidities for each individual (0, 1 or 2 or more) obtained from primary care records and supplemented with secondary care records in analyses conducted in the hospitalised cohort. Comorbidity count will incorporate conditions from the NHS Digital shielded patient list algorithm and others found to be associated with adverse COVID-19 outcomes. |
| Socioeconomic deprivation | An ordinal measure using quintiles of the Index of Multiple Deprivation (IMD), the official measure of relative deprivation for small areas in each country, based on residential postcode as recorded in HR or professional registration databases. Where residential postcode is not provided workplace postcode will be used as a surrogate. |

HR, human resource; NHS, National Health Service.

provide advice on study delivery and disseminate research findings. Representatives from ethnic minority healthcare professionals, including those who have contracted COVID-19, will also input into this stakeholder group. A Professional Expert Panel of doctors, nurses and ancillary staff from varying ethnic backgrounds and genders will meet bimonthly to provide advice and lived experience related to certain aspects of the project.

## STATISTICAL ANALYSIS PLAN
### Statistical principles
Reporting will be in line with Strengthening The Reporting of OBservational studies in Epidemiology (STROBE)[25] and REporting of studies Conducted using Observational Routinely collected health Data[26] guidelines. Any dissemination of study findings will follow best-practice guidelines for deductive disclosure. Only aggregate data will be included in publications, 95% CIs will be reported throughout and p-values<0.05 will indicate statistical significance. Multiple testing adjustments will not be made but emphasis will be on effect size rather than statistical significance. Deviations to this protocol and analysis plan will be outlined with justification in the final reporting of the results.

### Sample size
Sample size will be driven by the number of HCWs captured by the registration databases, with complete exposure information and successful linkage to healthcare data. The analysis plan is informed by likely event rates for each outcome, undertaking more sophisticated analyses for outcomes with sufficient power to provide meaningful results.

We will consider our largest primary data source (ESR) which captures 1.4 million NHS employees in England and Wales, for a conservative estimate of potential precision. The COVID-19 Infection Survey estimate 6.2% (95% CI: 5.0% to 7.6%) of adults in England would be seropositive for anti-SARS-CoV-2 antibodies.[27] Broadly assuming an equal infection rate in Wales, and that HCWs captured by the ESR represent the general population with a similar infection rate, we could expect to capture approximately 86 600 HCWs with history of COVID-19. A large proportion of the infected population will be asymptomatic, and a recent US study projected that 63% of infected cases went undiagnosed.[28] Assuming a similar diagnosis rate in England and Wales, we could expect to identify 32 042 COVID-19 cases. Of adults diagnosed with SARS-CoV-2 infections, the estimated hospitalisation rate is 25%[29] and mortality rate is 6.4%.[28] We could therefore expect to observe a minimum of 8000 hospitalisations and 2050 deaths resulting from COVID-19.

NHS workforce statistics data show 79.2% of the NHS workforce is White, 10.0% Asian and 6.1% Black. If associations between ethnicity and COVID-19 outcomes represent those observed in the general population, we could report differences between COVID-19 infection for Asian and Black ethnic categories compared with White with 0.1% and 0.2% precision, respectively. Similarly, in those with confirmed SARS-CoV-2 infection, we could report differences in COVID-19 mortality between Asian and Black ethnic categories, compared with White, with 0.9% and 0.8% precision respectively.

We will supplement the ESR with other HR databases and professional registration data sets, including SWISS, with an additional 170 000 HCWs in Scotland and the GMC, with an additional 336 000 GPs not captured by ESR. This increases the total number of HCWs incorporated into our analyses and should ensure sufficient numbers of outcome events.

### Missing data
Missing information for ethnicity, occupation, and other predictor variables will be reported and missing data patterns investigated. To improve completeness, we will use information from multiple data sources. Those with completely missing ethnicity information will be excluded from the main analysis. Comorbidity information will be identified using primary care records; thus, absence of comorbidity codes will be interpreted as absence of comorbidity (ie, not missing).

We will report the coverage of each analysis and the number (%) of HCWs included. Sensitivity to the

exclusion of missing data will be assessed using multiple imputation methods.

## Baseline characteristics

The frequency (%) of HCWs by ethnicity and occupational group will be reported. Core predictors (age, sex, comorbidity and deprivation) will be summarised by ethnicity and occupational group using frequency (%) for categorical measures and mean (SD) or median (IQR) for continuous. We will compare core predictors between those with recorded and missing ethnicity information.

For the HR substudy, additional ethnicity, occupation and exposure predictors will be summarised alongside the core predictors where available. Changes in roles, exposure and absences over time will be described across ethnicity category and occupational group.

All baseline data will be presented for the whole cohort and by country.

A flowchart, using the STROBE format, will present cohort selection for both the broad and HR substudy analyses, with exclusions explained. Average and overall follow-up time will be summarised for each cohort, by ethnicity, occupational group, and country.

## Outcome incidence

Adjusted annual and quarterly incidence rates, per 100 000 HCWs, for each outcome will be calculated for each country where available. Meta-analysis methods will combine the country-specific estimates to give an overall UK incidence rate estimate. Adjusted incidence rates by ethnicity and occupation will be calculated in the same way. All incidence rates will be adjusted for the four core predictors (age, sex, comorbidities and deprivation) using standardisation methods. The incidence rates of each outcome will be reported as the rate of cases in the final UK HCW cohort and within subgroups who have experienced preceding COVID-19 events (table 3: COVID-19 outcomes and definitions, identified via listed data sources, to be assessed in UK HCW cohort and within listed subgroups, via listed analysis approach in the planned study). The time between diagnosis and outcomes will be summarised and cumulative incidence curves will be plotted.

## Associations between ethnicity, occupation and COVID-19 outcomes

The relationship between ethnicity and COVID-19 outcomes will be assessed using Poisson, logistic and flexible parametric time-to-event regression models (table 3: COVID-19 outcomes and definitions, identified via listed data sources, to be asessed in UK HCW cohort and within listed subgroups, via listed analysis approach in the planned study). For each outcome, multivariable models will incorporate ethnic category and occupational group, and their interactions, adjusted for the four core predictors. The analyses will be conducted by country, and country-specific results combined using meta-analysis methods to produce UK estimates. We

will report incidence rate ratios, ORs or HRs with 95% CIs and p-values. Forest plots and cumulative incidence curves will be produced. Interactions between ethnicity and other predictors will be presented graphically to aid interpretation. Model assumptions will be assessed.

We will use directed acyclic graphs to map hypothesised causal associations between ethnicity, occupational group, the core predictors and COVID-19 outcomes. Mediation analysis methods will then be applied to disentangle the direct, indirect and mediated interactive effects of ethnic category and COVID-19 outcomes. The direct effects estimated using the approach are likely to be related to unmeasured explanatory factors, including experiences of structural discrimination and racism, rather than any genetic predispositions for differing outcomes.

## HR sub-study

We will conduct the analysis separately for all countries where both HR and outcomes data are available, country-specific results will be presented.

Each outcome will be evaluated using the same regression methods above. Multivariable models will incorporate ethnicity, occupation and the core predictors as before, while extended to incorporate additional predictor information. An ethnicity model will additionally adjust for religion, nationality and country of birth and incorporate their interactions with ethnicity, if appropriate. An occupation model will additionally adjust for primary area of work, role description, patient/non-patient facing, time in current role, place of work, pay grade and seniority. Finally, a COVID-19 exposure model will additionally adjust for absence information and total hours worked as time varying exposures, to assess the impact of changes in COVID-19 exposure over time. Forest plots and cumulative incidence curves will be produced. Interactions between ethnicity and other predictors, and the effects of time varying predictors, will be presented graphically to aid interpretation.

## Sensitivity and additional analyses

Dependent on the number of HCWs and events captured, we intend to conduct the following additional analyses:
► *Incorporating BMI and smoking into broad analysis*: adjust for BMI and smoking in COVID-19 outcome models if recorded by EHRs either during, or in the 24 months prior to, the study period.
► *Incorporating grouping by organisation*: incorporate clustering of HCWs within organisations by fitting hierarchical models to account for within organisation correlation and between organisation heterogeneity.
► *Granular analysis within the 'Asian/Asian British'* ethnicity category: investigate differences in COVID-19 outcomes between HCWs with South Asian (Indian, Pakistani and Bangladeshi), East Asian (Chinese) and Other Asian backgrounds.
► Granular analysis within the "medical staff" occupational group: investigate differences in COVID-19 outcomes between medical and dental subcategories.

- ► Subgroup analysis within hospital workers: investigate COVID-19 outcomes in hospital workers and cluster HCWs within hospitals using hierarchical models.
- ► *Subgroup analysis of ancillary workers*: investigate COVID-19 outcomes in ancillary workers through trusts that employ them directly.
- ► *Missing ethnicity*: explore approaches to deal with missing ethnicity information.
- ► *Missing data*: use multiple imputation to assess the robustness of our findings to missing data.

## ETHICS AND DISSEMINATION

Ethical approval for the UK-REACH programme has been obtained via the expedited HRA COVID-19 processes (REC ref: 20/HRA/4718, IRAS ID: 288316). A complimentary investigation into the ethical and legal implications of linking professional registration and healthcare data was commissioned in a parallel work package, the results of which are contained in a published policy report (https://uk-reach.org/main/publications/). In brief, the report considers how the relevant legal framework applies to UK-REACH, discusses the implications of carrying out large-scale data linkage and analysis in a trusted research environment and highlights the particular ethical issues arising in the context of research on HCWs' ethnicity and the COVID-19 pandemic. Ultimately, the report provides a framework for using and linking sensitive data in this project in a way that is ethically, legally and socially acceptable. The study outlined in this protocol is being guided by the relevant recommendations made within the policy report framework. All research data will be anonymised via the SAIL Databank before release to researchers. Study results will be submitted for publication in an open access peer-reviewed journals, disseminated through reports to Government and made available on our dedicated website (https://uk-reach.org/).

## DISCUSSION AND LIMITATIONS

As COVID-19 spreads, the number of cases requiring medical intervention will continue. It is crucial to maintain the healthcare system in the UK and control secondary transmission of COVID-19, and measures to identify and protect the HCWs most vulnerable to infection and adverse outcomes must be implemented. However, current COVID-19 risks in HCWs, particularly within those from historically disadvantaged populations, are poorly understood.

Published evidence of COVID-19 outcomes in HCWs are generally from small studies, in single centre or region settings outside the UK, which have not investigated associations with ethnicity.[30] COVID-19 studies which use observational data from non-random sampling, such as hospital admissions or voluntary participation, may be affected by collider bias.[31] One such study assesses outcomes using voluntary self-reported data from symptom trackers only,[32] thus results are at risk of selection bias due to reliance on accurate self-reporting of exposures and outcomes.

A recent study assessing risks of hospitalisation in HCWs and household members in Scotland[18] had insufficient numbers of HCWs from ethnic minority groups to reliably estimate associations between COVID-19 outcomes and ethnicity. This planned study is, to the authors' knowledge, the first UK-wide study to assess the relationship between ethnicity and COVID-19 outcomes in HCWs. This large study incorporates ambitious linkage of multiple anonymised HR, professional registration and NHS data sets providing UK-wide coverage of clinical and non-clinical HCWs. However, the lack of a single UK-wide database capturing all HCWs actively working limits our ability to identify all HCWs across all regions. We aim to identify non-clinical ancillary staff not directly employed by the NHS by incorporating contracting organisations.

The broad analysis will examine associations between ethnicity, occupational group and multiple important COVID-19 outcomes in HCWs employed in the UK, while adjusting for core predictors. Using readily available EHR data enables the investigation of complex models of public health resulting in scalable, population-based measurements of disease burden.[33] However, we foresee a trade-off between the scale and depth of the data, for example, while incorporating professional registration databases captures HCWs in non-NHS sectors, we are unable to identify whether they are actively working (not retired, on parental or long-term sick leave). Thus, a substudy will use detailed NHS HR databases to enable granular analyses of ethnicity, occupation and COVID-19 exposures in those actively working during the pandemic. Additional hypothesis generating analyses will be conducted where interesting results are found, for example if the study identified a higher mortality incidence in a particular ethnic group, a more detailed analysis investigating differences in disease severity by ethnic group could be planned.

Implications regarding the quality of the EHR data have been considered; multiple data sources will be utilised to limit the impact of missing information for the main exposures (ethnicity and occupation) and outcomes (diagnosis, hospitalisation, death). However, ethnicity information is notorious for being poorly recorded in EHR data and the missing information is unlikely to be missing at random. Even after taking steps to limit the impact of missing information, utilising a complete-case analysis approach may still introduce some sampling bias. Sensitivity analyses are planned to account for predictors expected to have high proportions of non-random missingness, such as ethnicity, BMI and smoking status. However, this study will be limited by the subset of COVID-19 cases that are diagnosed and recorded within the EHR data. Many HCWs with SARS-CoV-2 infections will not be diagnosed, with a US study projecting up to 63%.[28] Indeed, the study will not capture HCWs with mild or asymptomatic infections in settings where tests are restricted to those with symptoms, and results may be vulnerable to collider bias.[31] Incorporating both suspected and confirmed COVID-19 diagnoses is likely to

increase the number of infections captured, though it is also likely to reduce the specificity of diagnosis by incorporating false positive cases.

Findings from this study will directly inform evidence-based guidance to protect and address health inequalities in HCWs during the ongoing pandemic.

**Author affiliations**
¹Biostatistics Research Group, Department of Health Sciences, University of Leicester, Leicester, UK
²Genetic Epidemiology Research Group, Department of Health Sciences, University of Leicester, Leicester, UK
³Population Data Science, Swansea University Medical School, Swansea, UK
⁴Department of Respiratory Sciences, University of Leicester, Leicester, UK
⁵Department of Infection and HIV Medicine, University Hospitals of Leicester NHS Trust, Leicester, UK
⁶Institute of Health and Wellbeing, University of Glasgow, Glasgow, UK
⁷Diabetes Research Centre, University of Leicester, Leicester General Hospital, Leicester, UK
⁸Leicester Real World Evidence Unit, University of Leicester, Leicester General Hospital, Leicester, UK
⁹NIHR Leicester Biomedical Research Centre Respiratory Diseases, Glenfield Hospital, Leicester, UK
¹⁰Centre for Health Economics, University of York, York, UK

**Acknowledgements** This study makes use of anonymised data held in the Secure Anonymised Information Linkage (SAIL) Databank. We would like to acknowledge all the data providers who make anonymised data available for research. This work uses data provided by participants of the COVID-19 Symptoms Study, developed by ZOE Global Limited with scientific and clinical input from King's College London. We would also like to acknowledge all data providers who made anonymised data available for research. We wish to acknowledge the collaborative partnership that enabled acquisition and access to the deidentified data, which led to this output. The collaboration was led by BREATHE, the Health Data Research Hub for Respiratory Health, in partnership with SAIL Databank. We wish to acknowledge the input of ZOE Global Limited and King's College London in their development and sharing of the data, and their input into the understanding and contextualisation of data for COVID-19 research. All research conducted has been completed under the permission and approval of SAIL independent Information Governance Review Panel (IGRP) project number 1120.

**Collaborators** The UK-REACH Study Collaborative Group including: Manish Pareek, University of Leicester (Chief Investigator); Laura Gray, University of Leicester; Laura Nellums, University of Nottingham; Anna Guyatt, University of Leicester; Catherine Johns, University of Leicester; Chris McManus, University College London; Katherine Woolf, University College London; Ibrahim Akubakar, University College London; Amit Gupta, Oxford University Hospitals; Keith Abrams, University of York; Martin Tobin, University of Leicester; Louise Wain, University of Leicester; Sue Carr, University Hospital Leicester; Edward Dove, University of Edinburgh; Kamlesh Kunti, University of Leicester; David Ford, University of Swansea; Robert Free, University of Leicester.

**Contributors** MP is corresponding author for this manuscript and attests that all persons listed as authors meet the ICMJE criteria for authorship. Funding acquisition: CJ, DVF, KRA, KK, LG, MP and MT. Conceptualisation: CM, CO, DVF, KRA, KK, LG, LT and MP. Writing original draft: CM, CO, KRA, LG and LT. Revision and editing: CJ, CM, DVF, DM, KRA, KK, MP and MT. Data acquisition, analysis or interpretation: CM, CO, DVF, KRA, LG, LT and MP.

**Funding** This publication presents independent research supported by a grant from the MRC-UK Research and Innovation (MR/V027549/1) and the Department of Health and Social Care through the National Institute for Health Research (NIHR) rapid response panel to tackle COVID-19. Core funding was also provided by NIHR Biomedical Research Centres. The views expressed in this publication are those of the authors and not necessarily those of the National Health Service (NHS), the MRC, the NIHR or the Department of Health and Social Care. This work is carried out with the support of BREATHE—The Health Data Research Hub for Respiratory Health (MC_PC_19004) in partnership with SAIL Databank. BREATHE is funded through the UK Research and Innovation Industrial Strategy Challenge Fund and delivered through Health Data Research UK.

**Competing interests** LG leads the NIHR ARC EM Data2Health theme. KK is Director for the University of Leicester Centre for BME Health, Trustee of the South Asian Health Foundation, national NIHR ARC lead for Ethnicity and Diversity, and a member of Independent SAGE. He is supported by the NIHR ARC EM and the NIHR Leicester Biomedical Research Centre. CJ holds a Medical Research Council Clinical Research Training Fellowship (MR/P00167X/1).

**Patient and public involvement** Patients and/or the public were involved in the design, or conduct, or reporting or dissemination plans of this research. Refer to the Methods section for further details.

**Patient consent for publication** Not required.

**Provenance and peer review** Not commissioned; externally peer reviewed.

**ORCID iDs**
Lucy Teece http://orcid.org/0000-0001-6669-8534
Laura J Gray http://orcid.org/0000-0002-9284-9321
Carl Melbourne http://orcid.org/0000-0001-7216-4547
Chris Orton http://orcid.org/0000-0002-9561-2493
Christopher A Martin http://orcid.org/0000-0002-2337-4799
Kamlesh Khunti http://orcid.org/0000-0003-2343-7099
Martin Tobin http://orcid.org/0000-0002-3596-7874
Catherine John http://orcid.org/0000-0002-6057-2073
Keith R Abrams http://orcid.org/0000-0002-7557-1567
Manish Pareek http://orcid.org/0000-0003-1521-9964

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
