## [Reviewer comments · BMJ Open]

ARTICLE DETAILS

TITLE (PROVISIONAL)	United Kingdom Research study into Ethnicity And COVID-19 outcomes in Healthcare workers (UK-REACH): a retrospective cohort study using linked routinely collected data, study protocol
AUTHORS	Teece, Lucy; Gray, Laura; Melbourne, Carl; Orton, Chris; Ford, David; Martin, Christopher; McAllister, David; Khunti, Kamlesh; Tobin, Martin; John, Catherine; Abrams, Keith; Pareek, Manish

VERSION 1 – REVIEW

REVIEWER	van Gerwen, Maaïke Department of Otolaryngology- Head and Neck Surgery Institute for Translational Epidemiology Icahn School of Medicine at Mount Sinai New York
REVIEW RETURNED	08-Dec-2020

GENERAL COMMENTS	Teese et al. describe the study protocol for the United Kingdom Research study into ethnicity and COVID-19 outcomes in healthcare workers (UK-REACH), which is a retrospective cohort study linking data from anonymized human resource databases with professional registration and NHS datasets. Outcomes will be assessed between 01-02-2020 and date of data extraction, 30-09-2020 at a minimum to allow a minimum of 8 months follow-up. To overcome the limitation of the lack of one UK-wide database capturing all healthcare workers, investigators will actively reach out to identify as many healthcare workers as possible for inclusion. The authors provide a clear description of their study design, in- and exclusion criteria, primary predictor and outcomes. They provide details on the availability of study characteristics and protocols for dealing with missing variables and data. They clearly state that their start date for assessing COVID-19 outcomes is 01-02-2020 following the rationale that this is the date following the first confirmed COVID-19 case in UK. Data extraction date is 30-9-2020 at a minimum allowing 8 months follow-up. Can the authors provide additional details on the prospective timeline? Up until what date will data collection continue? Is there an end date for the proposed study?
--

REVIEWER	Golestaneh, Ladan Yeshiva University Albert Einstein College of Medicine, Medicine/Renal
REVIEW RETURNED	10-Dec-2020

GENERAL COMMENTS	In this manuscript Teece et al. propose to study the association of COVID-19 incidence and outcomes with race/ethnicity among
---

health workers registered in the UK. They plan to anonymously link several databases and will employ various sub-group analyses to evaluate the association of increased risk of incident COVID-19 infection and related mortality across all healthcare occupations. By focusing on health workers they have effectively removed the consideration of type of occupation (although there are many different degrees of exposure within the healthcare field which they try to account for) and possibly socio-economic status (presumably the participants are all employed by the NHS). I commend the authors for undertaking such a study and I find that these results are timely (the sooner the better) for us not only to better understand the socio-demographic risk factors for COVID outcomes, but also plan health policy as they pertain to protection of health care workers. The manuscript could be improved in the following manner:

1) In order for the manuscript to be clear to a general audience, one not familiar with the various UK databases, a brief primer on structure of NHS as the single payer health system sponsored by the government and the parity of salaries/job description would be helpful. A brief description of the databases used to ascertain outcomes are also warranted (page 6, lines 54-50; what is ICNARC).

2) In the abstract, the authors described doing a meta-analysis. A meta-analysis is a technique that pools data from multiple distinct publications to make an overall statement. A meta-analysis of what, or do they mean a pooled analysis of the various databases?

3) The authors propose mediation analysis to explore the association between ethnicity and Covid 19 outcomes. This would assume that race/ethnicity is somehow mediating the association via a genetic or innate predisposition independent of all other socio-economic, clinical, exposure-related factors. Is this in fact what the authors mean to convey?

4) The anonymized linkage procedure seems sound, but are some of the endpoints rare enough that potential identification of the individuals by colleagues may be possible? The authors are including all healthcare workers across various sectors increase the pool of participants which may help with this regard.

5) Unfortunately the numbers quoted on page 7, line 7 will likely need to be updated by the time of publication.

6) Why are the authors conducting a broad analysis as a first step? Why not focus their efforts on the second study design outlined on page 9, line 48. This proposal seems more suitable to the objectives defined as it allows for adjustment for occupational role and potential exposure.

7) How will linkage occur with the outcomes databases? The authors will send the relevant databases to SAIL?

8) By excluding those individuals without ethnicity data, is sampling bias introduced? There may be a race/ethnicity based differential inclination to self-identify race. Though unavoidable and a minor consideration, would make sure to mention it in the limitation, if it applies.

9) How will socio-economic deprivation be evaluated?

	10) Are the authors willing to consider "probable COVID infection" in those hospitalized with symptoms consistent with COVID but without a positive tests (false negative)? 11) The type of analytic method (time to outcome, cross-sectional) for incidence of COVID infection is not clear. Will the analysis be offset for time-period? Will there be adjustment for clustering? Does the "Analysis Approach" section on page 14 (line 45) apply to COVID incidence as well? Because it is stated as if measuring associations with COVID outcomes only. The authors do not discuss interaction testing. Will they be testing the interaction between race/ethnicity and type of occupation with regard to outcome (as written on page 11, line 17)?
--	---

VERSION 1 – AUTHOR RESPONSE

Reviewer 1: Dr. Maaïke van Gerwen

Teece et al. describe the study protocol for the United Kingdom Research study into ethnicity and COVID-19 outcomes in healthcare workers (UK-REACH), which is a retrospective cohort study linking data from anonymized human resource databases with professional registration and NHS datasets. Outcomes will be assessed between 01-02-2020 and date of data extraction, 30-09-2020 at a minimum to allow a minimum of 8 months follow-up. To overcome the limitation of the lack of one UK-wide database capturing all healthcare workers, investigators will actively reach out to identify as many healthcare workers as possible for inclusion.

Response: Thank you for summarizing the work.

The authors provide a clear description of their study design, in- and exclusion criteria, primary predictor and outcomes. They provide details on the availability of study characteristics and protocols for dealing with missing variables and data.

Response: Thank you for the positive comment.

They clearly state that their start date for assessing COVID-19 outcomes is 01-02-2020 following the rationale that this is the date following the first confirmed COVID-19 case in UK. Data extraction date is 30-9-2020 at a minimum allowing 8 months follow-up.

Response: Thank you for the positive comment.

Comment 1: *Can the authors provide additional details on the prospective timeline? Up until what date will data collection continue? Is there an end date for the proposed study?*

Response: UK-REACH has been funded as an Urgent Public Health study to August 2021 to provide expedited outputs that will be of direct relevance to the UK government. The cut-off date for collection of outcome data for this study will be the 31-01-2021 to give a maximum of 1-year follow-up, though we hope to gain permissions to adapt the project for use as a long-term resource.

To make the prospective timeline of the study clearer, we have removed all references to the “date of data extraction (30-09-2020)” and now refer to the “study end-date (31-01-2021)”, allowing a maximum of 1-year follow-up.

Reviewer 2: Dr. Ladan Golestaneh

In this manuscript Teece et al. propose to study the association of COVID-19 incidence and outcomes with race/ethnicity among health workers registered in the UK. They plan to anonymously link several databases and will employ various sub-group analyses to evaluate the association of increased risk of incident COVID-19 infection and related mortality across all healthcare occupations. By focusing on health workers they have effectively removed the consideration of type of occupation (although there are many different degrees of exposure within the healthcare field which they try to account for) and possibly socio-economic status (presumably the participants are all employed by the NHS). I commend the authors for undertaking such a study and I find that these results are timely (the sooner the better) for us not only to better understand the socio-demographic risk factors for COVID outcomes, but also plan health policy as they pertain to protection of health care workers.

Response: Thank you for seeing the value of our work and positive comment.

The manuscript could be improved in the following manner:

Comment 1: *In order for the manuscript to be clear to a general audience, one not familiar with the various UK databases, a brief primer on structure of NHS as the single payer health system sponsored by the government and the parity of salaries/job description would be helpful. A brief description of the databases used to ascertain outcomes are also warranted (page 6, lines 54-50; what is ICNARC).*

Response: Thank you for this comment. We agree that the article would be clearer to those unfamiliar with the NHS and UK database by providing descriptions of these.

We have now renamed the “Data sources” sub-section to “UK healthcare data sources”, and added a paragraph describing the NHS and its employment structures (page 6). In addition, we have expanded the “ICNARC” acronym to “Intensive Care National Audit & Research Centre data” and have added “SICSAG”, the equivalent dataset for Scotland (page 7).

Comment 2: *In the abstract, the authors described doing a meta-analysis. A meta-analysis is a technique that pools data from multiple distinct publications to make an overall statement. A meta-analysis of what, or do they mean a pooled analysis of the various databases?*

Response: Thank you for this comment. The planned analysis is to utilize meta-analysis methods to combine country-specific estimates from England, Scotland, Wales, and Northern Ireland. Part of the novelty of the study is bringing healthcare data from all four countries together, however due to the novelty we recognize the potential for data governance issues that might prevent us from conducting a complete pooled analysis. In planning to assess outcomes in each country separately, and then combine aggregate data using these methods, we are able to circumvent any potential data governance issues. Whilst we understand our planned analysis approach may be atypical, we believe it is appropriate for answering the study research questions.

We have updated the manuscript to further clarify that the meta-analysis will combine the country-specific estimates (Abstract – page 2; Outcome incidence – page 11; Analysis approach ~ renamed associations between ethnicity, occupation, and COVID-19 outcomes – page 11) and removed the reference to pooled analysis to better reflect the planned study (HR sub-study - page 12).

Comment 3: *The authors propose mediation analysis to explore the association between ethnicity and Covid 19 outcomes. This would assume that race/ethnicity is somehow mediating the association via a genetic or innate predisposition independent of all other socio-economic, clinical, exposure-related factors. Is this in fact what the authors mean to convey?*

Response: Thank you for this comment. We plan to undertake a mediation analysis to understand the extent of any associations found between ethnicity and poorer COVID-19 outcomes that are explained through associations with other factors. The mediation analysis will enable us to disentangle the indirect effects of ethnic category on COVID-19 outcomes (e.g. via socio-economic, clinical, comorbidities...) resulting in remaining “direct effects”. However, we do not believe there are any genetic predispositions within ethnic groups for COVID-19 outcomes. Rather, we understand ethnicity is a complex construct (Introduction, page 4) and as such, the “direct effect” of ethnicity are likely to be related to unmeasured explanatory factors, including experiences of structural discrimination and racism.

We have now added text in the manuscript to clarify this (Abstract – page 2; Analysis approach ~ renamed associations between ethnicity, occupation, and COVID-19 outcomes – page 11/12).

Comment 4: *The anonymized linkage procedure seems sound, but are some of the endpoints rare enough that potential identification of the individuals by colleagues may be possible? The authors are including all healthcare workers across various sectors increase the pool of participants which may help with this regard.*

Response: Thank you for this comment. We understand the importance and seriousness of protecting individual identities of those in the study and follow best-practice guidelines for deductive disclosure. Appropriate disclosure controls will be applied for all resulting publications; only aggregate data will be published and cell values and table designs will be assessed prior to extraction from SAIL.

We have added text in the manuscript to reflect this (Statistical principles – page 10).

Comment 5: *Unfortunately the numbers quoted on page 7, line 7 will likely need to be updated by the time of publication.*

Response: Thank you for highlighting this. We have updated the numbers reflect the latest situation (Introduction – page 4).

Comment 6: *Why are the authors conducting a broad analysis as a first step? Why not focus their efforts on the second study design outlined on page 9, line 48. This proposal seems more suitable to the objectives defined as it allows for adjustment for occupational role and potential exposure.*

Response: Thank you for your comment. The broad analysis is more inclusive as it captures healthcare workers not directly employed by the NHS, including General Practitioners and Dentists. The HR sub-study, while important as it allows examination of ethnicity and occupational exposure in greater granularity, misses these key groups of healthcare workers and will not include healthcare workers from Northern Ireland. As such, the results from the broad analysis will better represent the healthcare worker population in the UK as a whole, as well as having greater power to answer the primary aim of the study.

Comment 7: *How will linkage occur with the outcomes databases? The authors will send the relevant databases to SAIL?*

Response: Thank you for this comment. The NHS Wales Informatics Service (NWIS) will share information from the cohort of healthcare workers, including an anonymized study id number, with the relevant health outcome providers (including NHS Digital for England, Public Health Scotland for Scotland, and the Northern Ireland Honest Broker Service for Northern Ireland). These providers will conduct the linkage of the cohort to the outcome databases, and return de-identified outcome information with the anonymized study id number to SAIL. All identifiable health record data is kept entirely within the NHS, and is not accessible to the study analysis team. The linkage of outcome databases is described within the “Data linkage” subsection (page 7) and depicted the data flow diagram (Figure 1).

To make the linkage process clearer, we have updated and re-structured the “Data linkage” subsection (page 7).

Comment 8: *By excluding those individuals without ethnicity data, is sampling bias introduced? There may be a race/ethnicity based differential inclination to self-identify race. Though unavoidable and a minor consideration, would make sure to mention it in the limitation, if it applies.*

Response: Thank you for this comment. Ethnicity is likely to be inconsistently recorded across databases, thus we plan to utilise multiple data sources (including HR, registration, primary care, and secondary care records) to limit the amount of missing information. Those with ethnicity recorded as ‘not stated’ or with completely missing ethnicity information will be excluded from the main analyses. We agree that there are likely to be mechanism causing differences in samples between those who remain in the study and those who are excluded, and our planned approach may introduce some sampling bias.

We have incorporated text to address the limitations into the “Discussion and limitations” section (page 13) and have added additional plans for a sensitivity analysis to explore approaches to deal with missing ethnicity information (Sensitivity and additional analyses– page 12).

Comment 9: *How will socio-economic deprivation be evaluated?*

Response: Thank you for this comment. Socio-economic deprivation will be evaluated as an ordinal measure using quintiles of the Index of Multiple Deprivation, the official measure of relative deprivation for small areas in each country, based on residential postcode. We have provided details on the definition and measures for each of the core predictors in Table 4 (page 23).

To direct the reader to this table, and make it clearer that this information is there, we have added some text to the “Predictors” sub-section (page 9).

Comment 10: *Are the authors willing to consider "probable COVID infection" in those hospitalized with symptoms consistent with COVID but without a positive tests (false negative)?*

Response: Thank you for this comment. Three hospitalisation outcomes will be investigated within the study, specifically: all in-patient hospital admissions; COVID-19 hospitalisations with disease-specific codes or a positive test; COVID-19 admissions to intensive therapy units with disease-specific codes or a positive test. The disease-specific codes for the latter two outcomes include the ICD-10 code “U07.2: COVID-19, virus not identified” (Table 3 – page 21/22) which we believe would capture these patients. While we understand that combining confirmed and suspected COVID-19 hospitalisations may reduce the specificity of the outcome, we feel this action enables a more

consistent definition of hospitalization throughout the study (particularly in earlier months when testing was not so widely available) and allows for the presence of false-negative tests.

To direct the reader to this table, and make it clearer that this information is there, we have added some text to the “Outcomes” sub-section (page 9).

Comment 11: *The type of analytic method (time to outcome, cross-sectional) for incidence of COVID infection is not clear. Will the analysis be offset for time-period? Will there be adjustment for clustering? Does the "Analysis Approach" section on page 14 (line 45) apply to COVID incidence as well? Because it is stated as if measuring associations with COVID outcomes only. The authors do not discuss interaction testing. Will they be testing the interaction between race/ethnicity and type of occupation with regard to outcome (as written on page 11, line 17)?*

Response: Thank you for this comment. The incidence rates will be calculated as number of events observed overall, and by each quarter, divided by the size of the healthcare worker population. All incidence rates will be adjusted for age, sex, comorbidities, and deprivation using standardization methods. As well as reporting overall incidence rates, we will also report ethnicity-specific rates (incidence within each ethnic category) and occupational-specific rates (incidence within each occupational group). To address your questions in brief responses:

- The analytic method will be cross-sectional.
- The time-period will be quarterly.
- We are not adjusting for clustering in the main analysis as we will not have location information for all healthcare workers, only those directly employed by the NHS. We have planned a sensitivity analysis that will explore clustering by organization/hospital.
- The “analysis approach” section explains the planned analysis to explore associations between the exposures, predictors, and outcomes. These include COVID-19 diagnosis, hospitalization, and mortality.
- Yes, we will test the interaction between ethnic category and occupational group as written on page 11.

To make the analytic method clearer to the reader we have incorporated additional details into the “outcome incidence” subsection (page 11) and have renamed the “Analysis approach” subsection to “Associations between ethnicity, occupation, and COVID-19 outcomes” (page 11).

VERSION 2 – REVIEW

REVIEWER	van Gerwen, Maaïke Department of Otolaryngology- Head and Neck Surgery Institute for Translational Epidemiology Icahn School of Medicine at Mount Sinai New York
REVIEW RETURNED	04-Mar-2021

GENERAL COMMENTS	Dear author, Thank you for the opportunity to review your revised manuscript and for commenting on the suggested revisions. Kind regards, Maaïke van Gerwen
--

REVIEWER	Golestaneh, Ladan Yeshiva University Albert Einstein College of Medicine, Medicine/Renal
REVIEW RETURNED	11-Mar-2021
GENERAL COMMENTS	The authors have adequately addressed all previous concerns raised.